# Internal waves in marginally stable abyssal stratified flows

**Nikolay Makarenko**[1,2], **Janna Maltseva**[1,2], **Eugene Morozov**[3], **Roman Tarakanov**[3], and **Kseniya Ivanova**[4]

[1]Lavrentyev Institute of Hydrodynamics, 630090, Novosibirsk, Russia
[2]Novosibirsk State University, 630090, Novosibirsk, Russia
[3]Shirshov Institute of Oceanology, 117997, Moscow, Russia
[4]Aix-Marseille University, Marseille 06, France

**Correspondence:** N. I. Makarenko
(makarenko@hydro.nsc.ru)

**Abstract.**

The problem on internal waves in a weakly stratified two-layered fluid is studied semi-analytically. We discuss the 2.5-layer fluid flows with exponential stratification of both layers. The long-wave model describing travelling waves is constructed by means of scaling procedure with a small Boussinesq parameter. It is demonstrated that solitary wave regimes can be affected by the Kelvin — Helmholtz instability arising due to interfacial velocity shear in upstream flow.

## 1 Introduction

In this paper, we consider an analytical model of internal solitary waves in a two-layer fluid with the density continuously increasing with depth in both layers. This model is a development of non-linear two-layer models previously suggested by Ovsyannikov (1985), Miyata (1985) and Choi & Camassa (1999), as well as the latest 2.5-layer models considered by Voronovich (2003), Makarenko and Maltseva (2008, 2009a,b). Two-layer approximation is a standard model of sharp pycnocline in a stratified fluid with constant densities in each layer, but discontinuous at the interface. Correspondingly, the 2.5-layer model takes into account a slight density gradient in stratified layers which is comparable with the density jump at the interface. In all these cases, internal solitary waves can be described in closed form by the solutions resulting from the quadrature

$$\left(\frac{d\eta}{dx}\right)^2 = f(\eta) \tag{1}$$

for stationary wave elevation $\eta(x)$. The simplest version of non-linearity $f$ appears in a two-layer system; hence, it is the rational function $f(\eta) = P(\eta)/Q(\eta)$ where $P$ is a fourth degree polynomial, and $Q$ depends linearly on $\eta$. Equation (1) also appears as a travelling wave equation for non-linear evolution systems being similar to single-layer dispersive Green – Naghdi model (see Choi & Camassa, 1999). These non-linear dispersive equations can be obtained by means of long-wave perturbation technique as well as by the Whitham's variational method. Several authors noted that solitary wave solutions of such approximate models are in good agreement with the numerical solutions of fully non-linear Euler equations for a perfect two-layer fluid. In this context, Camassa & Tiron (2011) also compared numerical travelling wave-solutions, supported by smooth stratification, with the known explicit solitary-wave solutions in order to optimize a two-layer model of the Euler system with smooth stratification.

We apply the method of derivation involving asymptotic analysis of the non-linear Dubreil-Jacotin — Long equation that results from fully nonlinear Euler equations of stratified fluid. Long-wave scaling procedure uses a small Boussinesq parameter which characterizes slightly increasing density in the layers and a small density jump at their interface. This method combines the approaches applied formerly to a pure two-fluid system with perturbation technique discussed for the first time by Long (1965) and developed by Benney and Ko (1978) for a continuous stratification. Parametric range of solitary wave is considered in the framework of the constructed mathematical model. It is demonstrated that these wave regimes can approach the parametric domain of the Kelvin — Helmholtz instability. The stability of solitary travelling-wave solutions of the Euler equations for continuously stratified, near two-layer fluids was studied numerically and analytically by Almgren, Camassa, & Tiron (2012).

They demonstrated that the wave-induced shear can locally reach unstable configurations and give rise to local convective instability. This is in good qualitative agreement with the laboratory experiments performed by Grue et al. (2000). It seems that such a marginal stability of long internal waves could explain the formation mechanism of a very long billow trains in abyssal flows observed by van Haren et al. (2014).

## 2   Basic Equations

We consider a 2D motion of inviscid two-layer fluid which is weakly stratified due to gravity in both layers. The fully nonlinear Euler equations describing the flow are

$$\rho(u_t + uu_x + vu_y) + p_x = 0, \tag{2}$$

$$\rho(v_t + uv_x + vv_y) + p_y = -\rho g, \tag{3}$$

$$\rho_t + u\rho_x + v\rho_y = 0, \tag{4}$$

$$u_x + v_y = 0, \tag{5}$$

where $\rho$ is the fluid density, $(u,v)$ is the fluid velocity, $p$ is the pressure and $g$ is the gravity acceleration. We assume that the flow domain is bounded by the flat bottom $y = -h_1$ and the rigid lid $y = h_2$ (see Fig. 1), with the boundary condition

$$v = 0\big|_{y=-h_1,\, y=h_2}. \tag{6}$$

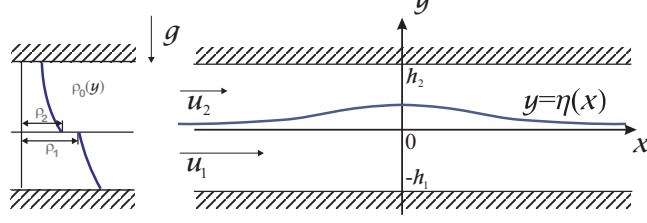

**Figure 1.** Scheme of the flow

The layers are separated by the interface $y = \eta(x,t)$ with the equilibrium level at $y = 0$. Non-linear kinematic and dynamic boundary conditions at this interface are

$$\eta_t + u\eta_x = v\big|_{y=\eta}, \quad [p] = 0\big|_{y=\eta} \tag{7}$$

where the square brackets denote the discontinuity jump at the interface between the layers. Non-disturbed parallel flow has no vertical velocity and elevation (i.e. $v = 0$, $\eta = 0$) but the horizontal velocity $u = u_0(y)$ may be piece-wise constant,

$$u_0(y) = \begin{cases} u_1 & (-h_1 < y < 0), \\ u_2 & (0 < y < h_2). \end{cases} \tag{8}$$

In this stationary case, the fluid density $\rho = \rho_0(y)$ and pressure $p = p_0(y)$ should be coupled by the hydrostatic equation $dp_0/dy = g\rho_0$. We consider the density profile depending exponentially on height,

$$\rho_0(y) = \begin{cases} \rho_1 \exp\left(-N_1^2 y/g\right) & (-h_1 < y < 0), \\ \rho_2 \exp\left(-N_2^2 y/g\right) & (0 < y < h_2), \end{cases} \tag{9}$$

where $N_j = \text{const}$ is the Brunt — Väisälä frequency in the $j$-th layer, and constant densities $\rho_1$ and $\rho_2$ are related as $\rho_2 < \rho_1$. The special case $N_j = 0$ $(j = 1,2)$ gives a familiar two-fluid system with piece-wise constant density $\rho = \rho_j$ in the $j$-th layer.

Further we consider a steady non-uniform flow, hence we have $\eta_t = 0$ and $u_t = v_t = \rho_t = 0$ in Eqs. (2) – (4). We introduce the stream function $\psi$ by standard formulae $u = \psi_y$, $v = -\psi_x$, hence the mass conservation implies the dependence $\rho = \rho(\psi)$, and pressure $p$ can be found from the Bernoulli equation

$$\frac{1}{2}|\nabla\psi|^2 + \frac{1}{\rho(\psi)}p + gy = b(\psi). \tag{10}$$

Seeking for a solitary-wave solutions, we require that the upstream velocity of the fluid $(u,v)$ tends to $(u_j,0)$ as $x \to -\infty$. In this case, boundary conditions (6) transform to the conditions for the stream function as

$$\psi = -u_1 h_1\big|_{y=-h_1}, \; \psi = 0\big|_{y=\eta}, \quad \psi = u_2 h_2\big|_{y=h_2}. \tag{11}$$

It is known (Yih, 1980) that system (2) – (5) can be reduced in a stationary case to the non-linear Dubreil-Jacotin — Long (DJL) equation for the stream function

$$\rho(\psi)\nabla^2\psi + \rho'(\psi)\left(gy + \frac{1}{2}|\nabla\psi|^2\right) = H'(\psi). \tag{12}$$

Here, the function $H(\psi) = \rho(\psi)b(\psi)$ involves the Bernoulli function $b(\psi)$ and the density function $\rho(\psi)$, so that $H$ is specified by the upstream condition. More exactly, the density function is determined by the relation $\rho(\psi) = \rho_0(\psi/u_j)$ in the $j$-th layer, and the Bernoulli function $b(\psi)$ is defined by the formula

$$b = \begin{cases} \dfrac{1}{2}u_1^2 + g\dfrac{\psi}{u_1} + \dfrac{g^2}{N_1^2}\left(1 - e^{\frac{N_1^2\psi}{gu_1}}\right) & (-h_1 < y < \eta(x)), \\[3mm] \dfrac{1}{2}u_2^2 + g\dfrac{\psi}{u_2} + \dfrac{g^2}{N_2^2}\left(1 - e^{\frac{N_2^2\psi}{gu_2}}\right) & (\eta(x) < y < h_2). \end{cases}$$

As a consequence, we can rewrite the DJL equation (12) as follows:

$$\nabla^2\psi = \frac{N_j^2}{gu_j}\left\{g\left(y - \frac{\psi}{u_j}\right) + \frac{1}{2}\left(|\nabla\psi|^2 - u_j^2\right)\right\}, \tag{13}$$

where $j = 1$ is related to the lower layer, and $j = 2$ to the upper layer. Further, in accordance with relations (7) and (10),

the continuity of pressure $p$ provides non-linear boundary condition for stream function $\psi$

$$\left[\rho(\psi)(|\nabla\psi|^2 + 2gy - 2b(\psi))\right] = 0\big|_{y=\eta}. \tag{14}$$

Using the explicit form of functions $\rho(\psi)$ and $b(\psi)$, condition (14) can be also rewritten in detail as follows:

$$2g(\rho_1 - \rho_2)\eta =$$
$$= \rho_2\left(|\nabla\psi|^2 - u_2^2\right)\big|_{y=\eta(x)+0} - \rho_1\left(|\nabla\psi|^2 - u_1^2\right)\big|_{y=\eta(x)-0}.$$

We reformulate this boundary condition in view of conservation of the total horizontal momentum in a steady two-layer flow, which has integral formulation

$$\int_{-h_1}^{h_2} \left(p + \rho u^2\right) dy = C$$

where constant $C$ is determined by the upstream condition. Excluding pressure $p$ from this relation using the Bernoulli equation (10) leads to the integral relation

$$\rho_1 \int_{-h_1}^{\eta(x)} e^{-\frac{N_1^2\psi}{gu_1}} \Psi_1 \, dy + \rho_2 \int_{\eta(x)}^{h_2} e^{-\frac{N_2^2\psi}{gu_2}} \Psi_2 \, dy = C \tag{15}$$

where the integrand functions $\Psi_j$ are

$$\Psi_j = \psi_y^2 - \psi_x^2 + u_j^2 + 2g\left(\frac{\psi}{u_j} - y\right) - \frac{2g^2}{N_j^2}\left(e^{\frac{N_j^2\psi}{gu_j}} - 1\right),$$

and constant $C$ depends on the parameters of the upstream flow as follows:

$$C = 2\rho_1 g\left[\left(e^{\frac{N_1^2 h_1}{g}} - 1\right)\left(\frac{u_1^2}{N_1^2} + \frac{g^2}{N_1^4}\right) - \frac{gh_1}{N_1^2}\right] +$$
$$+ 2\rho_2 g\left[\left(1 - e^{-\frac{N_2^2 h_2}{g}}\right)\left(\frac{u_2^2}{N_2^2} + \frac{g^2}{N_2^4}\right) - \frac{gh_2}{N_2^2}\right].$$

It is important here that the integral relation (15) is equivalent to the boundary condition (14) which is rather simple. This equivalence can be checked immediately by differentiation the relation (15) with respect to the variable $x$, so the integrals can be evaluated explicitly due to Eq.(13). Equation (15) will be used later instead of (14) by the construction model differential equation for the function $\eta(x)$ describing strongly nonlinear waves.

## 3 Non-Dimensional Formulation

Now we introduce scaled independent variables $\bar{x}$, $\bar{y}$ and scaled unknown functions $\bar{\eta}$, $\bar{\psi}$ in order to reformulate the basic equations in the dimensionless form. Namely, the fixed ratio $h_1/\pi$ is used as an appropriate length scale for $x, y, \eta$,

and normalized volume discharges $u_j h_j/\pi$ serve as the units for the stream function; thus, we have

$$(x, y, \eta) = \frac{h_1}{\pi}(\bar{x}, \bar{y}, \bar{\eta}), \quad \psi = \frac{u_j h_j}{\pi}\bar{\psi}$$

separately in the lower layer ($j = 1$) or in the upper layer ($j = 2$). The number $\pi$ is only introduced here due to the specific form of trigonometric modal functions which are typical for the exponential density (9). Scaling procedure with this density profile uses the Boussinesq parameters $\sigma_1$, $\sigma_2$ and the Atwood number $\mu$ defined by the formulae

$$\sigma_j = \frac{N_j^2 h_j}{\pi g} \quad (j = 1, 2), \qquad \mu = \frac{\rho_1 - \rho_2}{\rho_2}. \tag{16}$$

Here, constants $\sigma_j$ characterize the slope of the density profile in continuously stratified layers, and parameter $\mu$ determines the density jump at interface.

Following Turner (1973), we introduce densimetric (or internal) Froude number

$$F_j = \frac{u_j}{\sqrt{g_j h_j}} \qquad (j = 1, 2)$$

which presents scaled fluid velocity $u_j$ in the $j$-th layer, defined with reduced gravity acceleration $g_j = (\rho_1 - \rho_2)g/\rho_j$. In addition to the Froude numbers $F_j$, it is also convenient to use the pair of the Long's numbers $\lambda_j$ given by the formula

$$\lambda_j = \frac{N_j h_j}{\pi u_j} \quad (j = 1, 2).$$

The Long's numbers $\lambda_j$ are coupled with the Boussinesq parameters $\sigma_1$, $\sigma_2$, the Atwood number $\mu$ and the Froude numbers $F_j$ by the relations

$$\lambda_1^2 = \frac{\pi\sigma_1(1 + \mu)}{\mu F_1^2}, \quad \lambda_2^2 = \frac{\pi\sigma_2}{\mu F_2^2}. \tag{17}$$

Finally, we introduce the ratio of undisturbed thicknesses of the layers $r = h_1/h_2$. By that notation, we locate the bottom as $\bar{y} = -\pi$, and relation $\bar{y} = \pi/r$ defines the rigid lid. Thus, we obtain the equations for scaled stream function $\bar{\psi}$ and non-dimensional wave elevation $\bar{\eta}$ as follows (bar is omitted throughout what follows):

$$\nabla^2\psi + \lambda_1^2(\psi - y) = \frac{1}{2}\sigma_1\left(|\nabla\psi|^2 - 1\right) \tag{18}$$

in the lower layer $-\pi < y < \eta(x)$, and

$$\nabla^2\psi + \lambda_2^2 r^2(\psi - ry) = \frac{1}{2}\sigma_2\left(|\nabla\psi|^2 - r^2\right) \tag{19}$$

in the upper layer $\eta(x) < y < \pi/r$. Kinematic boundary conditions (11) can be rewritten now as follows:

$$\psi(x, -\pi) = -\pi, \quad \psi(x, \eta(x)) = 0, \quad \psi(x, \pi/r) = \pi. \tag{20}$$

Correspondingly, Eq. (14) providing continuity of pressure at interface $y = \eta(x)$ leads to nonlinear boundary condition

$$2\eta = F_2^2(|\nabla\psi|^2 - r^2)\big|_{y=\eta+0} - F_1^2(|\nabla\psi|^2 - 1)\big|_{y=\eta-0}, \quad (21)$$

and the dimensionless version of integral relation (15) takes the form

$$\int_{-\pi}^{\eta} e^{-\sigma_1\psi}\,\Psi_1\,dy + \int_{\eta}^{\pi/r} e^{-\sigma_2\psi}\,\Psi_2\,dy = C \quad (22)$$

where is denoted

$$\Psi_1 = \frac{\mu F_1^2}{2}\left(\psi_y^2 - \psi_x^2 + 1\right) + \frac{1+\mu}{\pi}\left(\psi - y - \frac{e^{\sigma_1\psi}-1}{\sigma_1}\right),$$

$$\Psi_2 = \frac{\mu F_2^2}{2r^3}\left(\psi_y^2 - \psi_x^2 + r^2\right) + \frac{1}{\pi r}\left(\psi - ry - \frac{e^{\sigma_2\psi}-1}{\sigma_2}\right).$$

Constant

$$C = \pi\mu\left(F_1^2 + \frac{F_2^2}{r^2}\right) +$$

$$+(1+\mu)\frac{e^{\sigma_1\pi}-1-\sigma_1\pi}{\pi(\lambda_1^2+\sigma_1^2)} + \frac{1-\sigma_2\pi - e^{-\sigma_2\pi}}{\pi r^2(\lambda_2^2+\sigma_2^2)}$$

is chosen here so that the horizontal upstream flow given by the solution

$$\eta = 0, \quad \psi_0(y) = \begin{cases} y & (-\pi < y < 0), \\ ry & (0 < y < \pi/r) \end{cases} \quad (23)$$

satisfies momentum relation (22).

The model of fully nonlinear travelling waves in a two-layer irrotational flows, with the interface $y = \eta(x)$ between the fluids with constant densities $\rho_2$ in the upper layer and $\rho_1 > \rho_2$ in the lower layer, can be specified as follows. In this limit case, at least formally, the Boussinesq parameters $\sigma_j$ and Long's numbers $\lambda_j$ vanish: $\sigma_1 = \sigma_2 = \lambda_1 = \lambda_2 = 0$. Therefore, we obtain the Laplace equation

$$\nabla^2\psi = 0 \quad (24)$$

instead of Eqs. (18)–(19), but all the boundary conditions (20) and (21) still remain unchanged.

## 4  Spectrum of Harmonic Waves

In many cases, parametric range of solitary waves can be determined *a priori* as the domain being supercritical with respect to the spectrum of small–amplitude sinusoidal waves. It is helpful while the critical phase speed can be simply defined from the dispersion relation of infinitesimal waves. In our case, linearizing of Eqs. (18)–(21) for the upstream solution (23) leads to the dispersion relation

$$\Delta(k; F_1, F_2) = 0 \quad (25)$$

for stationary harmonic wave-packets

$$\eta(x) = a\,e^{ikx}, \quad \psi = \psi_0(y) + W(y)\,e^{ikx}.$$

Here, $k$ is the non-dimensional wave-number, $a$ is the amplitude of interfacial wave, and $W(y)$ is the modal eigenfunction which describes deformation of streamlines within the fluid layers. For the given Long's numbers $\lambda_1$, $\lambda_2$ and the Boussinesq parameters $\sigma_1$, $\sigma_2$, we also introduce non-dimensional values

$$\varkappa_j = \sqrt{|\lambda_j^2 - k_j^2 - \frac{1}{4}\pi^2\sigma_j^2|} \qquad (j = 1,2), \quad (26)$$

where $k_1 = rk$ and $k_2 = k$ are dimensionless wave-numbers specified for each layer. According to these notations, dispersion function $\Delta(k; F_1, F_2)$ in (25) has the form

$$\Delta = F_1^2\left(\varkappa_1\mathrm{Cot}_1\varkappa_1 + \frac{\pi\sigma_1}{2}\right) + F_2^2\left(\varkappa_2\mathrm{Cot}_2\varkappa_2 - \frac{\pi\sigma_2}{2}\right) - 1$$

where functions $\mathrm{Cot}_j$ $(j = 1,2)$ are denoted as follows:

$$\mathrm{Cot}_j\,\varkappa_j = \begin{cases} \cot\varkappa_j & (\lambda_j^2 > k_j^2 + \frac{1}{4}\pi^2\sigma_j^2) \\[2mm] \coth\varkappa_j & (\lambda_j^2 < k_j^2 + \frac{1}{4}\pi^2\sigma_j^2). \end{cases}$$

In fact, function $\Delta$ takes such a combined form since modal function $W(y)$ depends on $y$ trigonometrically or hyperbolically, if the radicand term $\lambda_j^2 - k_j^2 - \frac{1}{4}\pi^2\sigma_j^2$ in (26) is positive or negative. Explicit formulae for these modal eigenfunctions $W(y)$ are given in Appendix A.

Spectrum of stationary harmonic waves, defined on the $(F_1, F_2)$-plane, is formed by the Froude points $(F_1, F_2)$ so that dispersion function $\Delta(k; F_1, F_2)$, which is even in $k$, has at least one pair of real roots $\pm k$. Wave modes differ by the number of these pairs, and this number can change only by passing of the root across the value $k = 0$. Therefore, the modal bounds should satisfy the equation $\Delta(0; F_1, F_2) = 0$; these bounds are defined by separate branches of the curve

$$F_1^2\left\{\sqrt{\lambda_1^2 - \left(\frac{\pi\sigma_1}{2}\right)^2}\cot\sqrt{\lambda_1^2 - \left(\frac{\pi\sigma_1}{2}\right)^2} - \frac{\pi\sigma_1}{2}\right\} + \quad (27)$$

$$+F_2^2\left\{\sqrt{\lambda_2^2 - \left(\frac{\pi\sigma_2}{2}\right)^2}\cot\sqrt{\lambda_2^2 - \left(\frac{\pi\sigma_2}{2}\right)^2} + \frac{\pi\sigma_2}{2}\right\} = 1$$

where parameters $\lambda_j$ should be coupled with the Froude numbers $F_j$ using the formulae (17).

We emphasize that parameters $\sigma_j$ characterize the slope of density profile in continuously stratified layers, and $\mu$ defines the density jump at the interface. As usual, all these parameters are small in the case of low stratification. However, the interfacial mode dominates over the modes of internal waves in stratified layers when $\sigma_j \ll \mu$ is valid. In this

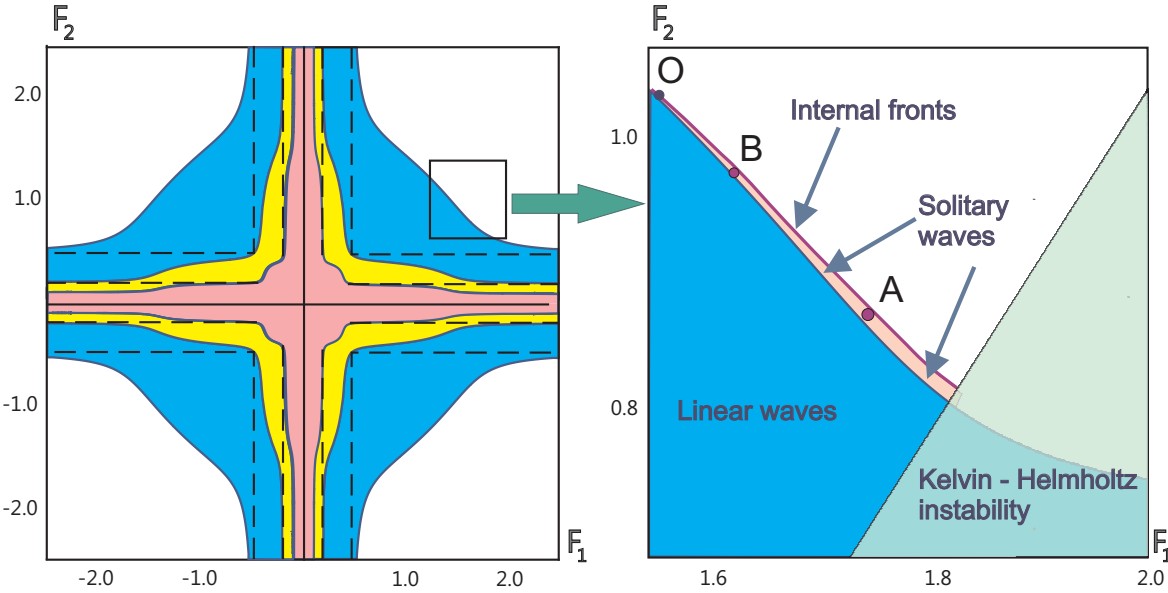

**Figure 2.** Spectrum of linear waves (colored modes 1-3) (left) and fragment of parametric domain of solitary waves (right).

limit case, linearized boundary conditions (20)–(21), considered together with the linear Laplace equation (24), lead to the standard dispersion relation of two-layer fluid

$$F_1^2 \, rk \coth rk + F_2^2 \, k \coth k = 1. \tag{28}$$

This relation determines only a single pair of real wavenumbers of the interfacial mode, so the spectral domain of a perfect 2-layer system occupies the unit disk

$$F_1^2 + F_2^2 \leqslant 1. \tag{29}$$

The 2.5-layer model starts with the hypotheses that the Boussinesq parameters $\sigma_1$, $\sigma_2$ and the Atwood number $\mu$ are of the same order, so we can use a single small parameter $\sigma$ by setting

$$\sigma = \sigma_1 = \sigma_2 = \mu. \tag{30}$$

The limit passage $\sigma \to 0$ is singular because the Long's numbers $\lambda_j$ involve the ratios $\sigma_j/\mu$ in formulae (17). However, condition (30) allows us to simplify the spectral portrait; hence modal curve (27) defining the critical wave speeds takes the form

$$\sqrt{\pi} F_1 \cot \frac{\sqrt{\pi}}{F_1} + \sqrt{\pi} F_2 \cot \frac{\sqrt{\pi}}{F_2} = 1. \tag{31}$$

Figure 2 demonstrates the parts of the spectrum defined by curve (31) for the dominating modes. The domain covered only by the first mode is marked with the blue color. Correspondingly, the embedded domain of the second mode is highlighted with the yellow, and the third mode is marked with the pink color. It is important that this spectrum differs essentially from the ordinary 2-layer spectrum (29), even the flow is characterized with a pair of the Froude numbers $F_1$, $F_2$, defined by the same manner. We specially note that the 2.5-layer spectrum extents infinitely on the spectral plane by involving unbounded Froude numbers $F_j$.

## 5  The Non-Linear Long-Wave Model

The derivation procedure of non-linear long-wave 2.5-layer model should involve, in accordance with hypothesis (30), the slow horizontal variable $\xi = \sqrt{\sigma}\, x$, as it was demonstrated by Benney & Ko (1978) in the case of slight linear stratification. Scaling with the parameter $\sigma$ gives the equation

$$\sigma \psi_{\xi\xi} + \psi_{yy} + \lambda_1^2 \left(\psi - y\right) = \frac{1}{2}\sigma \left(\sigma \psi_\xi^2 + \psi_y^2 - 1\right) \tag{32}$$

in the lower layer $-\pi < y < \eta(\xi)$, and

$$\sigma \psi_{\xi\xi} + \psi_{yy} + \lambda_2^2 r^2 \left(\psi - y\right) = \frac{1}{2}\sigma \left(\sigma \psi_\xi^2 + \psi_y^2 - r^2\right) \tag{33}$$

in the upper layer $\eta(\xi) < y < \pi/r$. Kinematic boundary conditions (11) can be rewritten now as follows:

$$\psi(\xi, -\pi) = -\pi, \quad \psi(\xi, \eta(\xi)) = 0, \quad \psi(\xi, \pi/r) = \pi. \tag{34}$$

We find that stream function $\psi$ is expanded in a power series with respect to $\sigma$ as

$$\psi = \psi^{(0)}(\xi, y) + \sigma \psi^{(1)}(\xi, y) + ... \tag{35}$$

where the leading-order term $\psi^{(0)}$ defines the hydrostatic mode, and the coefficient $\psi^{(1)}$ provides the correction due to non-linear dispersion. All these coefficients $\psi^{(k)}$ can be uniquely determined from equations (32) and (33) (with fixed Long's numbers $\lambda_1$ and $\lambda_2$) under kinematic boundary condition (34). Thus, we obtain

$$\psi^{(0)} = y - \eta \frac{\sin \alpha_1(y)}{\sin \alpha_1(\eta)} \qquad (-\pi < y < \eta),$$

and

$$\psi^{(0)} = ry - r\eta \frac{\sin \alpha_2(y)}{\sin \alpha_2(\eta)} \qquad (\eta < y < \pi/r),$$

where

$$\alpha_1(y) = \lambda_1(\pi + y), \qquad \alpha_2(y) = \lambda_2(\pi - ry). \tag{36}$$

The final form of dispersive term $\psi^{(1)}$ is much more complicated, therefore this coefficient is given in Appendix A.

Now we substitute power expansion (35) for function $\psi$ into the scaled version of integral relation (22) and truncate the terms with the powers higher than the first power of $\sigma$. By that, equation (22) reduces to the first-order ordinary differential equation for the wave elevation $\eta(x)$ and is written as

$$\left( \frac{d\eta}{dx} \right)^2 = \eta^2 \frac{D(\eta; F_1, F_2)}{Q(\eta; F_1, F_2)}. \tag{37}$$

Here function $D$ is given by the formula

$$D(\eta; F_1, F_2) =$$

$$= \sqrt{\pi} F_1 \cot \alpha_1(\eta) + \sqrt{\pi} F_2 \cot \alpha_2(\eta) + \frac{1}{3}(1 - r)\eta - 1$$

where $\alpha_1$ and $\alpha_2$ should be taken as

$$\alpha_1(\eta) = \frac{\pi + \eta}{\sqrt{\pi} F_1}, \quad \alpha_2(\eta) = \frac{\pi - r\eta}{\sqrt{\pi} F_2} \tag{38}$$

since we have at the leading order in $\sigma$ the relations $\lambda_j = 1/\sqrt{\pi} F_j$ $(j = 1, 2)$ obtained under condition (30). Denominator $Q$ in (37) has a complicated form, therefore this function is given in Appendix C. Solitary-wave solutions of Eq. (37) are given in the implicit form by the formula

$$x = \pm \int_a^\eta \sqrt{\frac{Q(s; F_1, F_2)}{D(s; F_1, F_2)}} \frac{ds}{s} \tag{39}$$

where parameter $a$ determines non-dimensional amplitude of the wave.

Small-amplitude waves can be modelled by simplified weakly nonlinear version of the Eq.(37) having the form

$$\left( \frac{d\eta}{dx} \right)^2 = \eta^2 \frac{D_0 + D_1 \eta + D_2 \eta^2}{Q(0; F_1, F_2)} \tag{40}$$

where the coefficients $D_0 = D(0; F_1, F_2)$ and $D_1 = D'_\eta(0; F_1, F_2)$ are

$$D_0 = \sqrt{\pi} F_1 \cot \frac{\sqrt{\pi}}{F_1} + \sqrt{\pi} F_2 \cot \frac{\sqrt{\pi}}{F_2} - 1,$$

$$D_1 = -\cot^2 \frac{\sqrt{\pi}}{F_1} + r \cot^2 \frac{\sqrt{\pi}}{F_2} + \frac{2}{3}(r - 1),$$

and the explicit form of coefficient $D_2$ is not important here. This model takes into account the balance of quadratic and cubic nonlinearities in the weakly-nonlinear KdV–mKdV – Gardner model (Kakutani and Yamasaki 1978; Gear and Grimshaw, 1983; Helfrich and Melville 2006; Grimshaw et al 2002).

Solitary wave regimes are obtained depending on the multiplicity of the roots $a_j(F_1, F_2, r)$ $(j = 1, 2)$ of the numerator on the right-hand side of (40). Profile of solitary wave is given by the formula

$$\eta(x) = a \frac{1 - \tanh^2 kx}{1 - \theta^2 \tanh^2 kx}, \quad k = \frac{a\sqrt{3/q_*}}{2\theta},$$

with $q_* = Q(0)$, $a = a_1$ and $\theta^2 = a_1/a_2 < 1$, and the bore (internal front) corresponds to the double root $a = a_1 = a_2$, it has the following profile

$$\eta(x) = \frac{a}{2} \left( 1 + \tanh kx \right), \quad k = \frac{a\sqrt{3/q_*}}{2}.$$

Parametric range of strongly nonlinear solitary waves described by Eq. (37) is formed by the domain in $(F_1, F_2)$-plane where the radical function $Q/D$ in (39) is ensured to be non-negative. It is easy to check that $Q(0; F_1, F_2) > 0$; hence function $Q(s; F_1, F_2)$ is positive in the vicinity of point $s = 0$. Therefore, function $D$ plays the determining role here. Depending on $F_1$ and $F_2$, this function can change the sign even by small $s$, where the leading-order coefficient $D_0$ from formula (40) dominates. As a consequence, the map of solitary-wave regimes is formed by the Froude numbers $(F_1, F_2)$ such that inequality $D_0(F_1, F_2) > 0$ holds. Indeed, this inequality defines the range of non-linear waves, which are supercritical with respect to the phase speed of linear harmonic wave-packets (see Fig. 2).

## 6 Marginally Stable Layered Flows

Large-amplitude internal waves are generated in deep ocean layers due to the interaction of internal tides with irregular bottom topography near underwater ridges (Morozov 1995, Morozov et al. 2010, Morozov 2018). These waves play a significant role in the energy transformation and mass transport in the oceanic stratified flows while they intensify mixing of the abyssal waters. Note that internal Froude numbers $F_1$ and $F_2$ characterize the magnitude of the velocity

jump at the interface in upstream flow. The shear $u_1 \neq u_2$ between the layers can initiate the development of the Kelvin — Helmholtz instability which provides non-stationary formation of billow trains (Thorpe 1985; Drazin 2002). In this context, long-wave perturbations give the greatest contribution to this instability due to their increased power intensity. Constant two-layer flow is linearly stable in the long-wave limit if the inequality

$$|u_1 - u_2| < \sqrt{\frac{g(\rho_1 - \rho_2)(\rho_1 h_2 + \rho_2 h_1)}{\rho_1 \rho_2}} \qquad (41)$$

holds, and this flow is unstable in the opposite case (Lamb 1932). Exactly the same bound (41) follows from the *non-linear* stability criteria predicted by the shallow water theory (Ovsyannikov 1979; see also Baines 1995; Gavrilyuk, Makarenko, Sukhinin 2017) for a *variable* difference $|\bar{u}_1 - \bar{u}_2|$ and *variable* layer thicknesses

$$\bar{h}_1(x,t) = h_1 + \eta(x,t), \quad \bar{h}_2(x,t) = h_2 - \eta(x,t).$$

In this case, velocity $\bar{u}_j$ is the horizontal fluid velocity averaged over the depth of the j-th layer (j=1,2),

$$\bar{u}_1(x,t) = \frac{1}{\bar{h}_1(x,t)} \int\limits_{-h_1}^{\eta(x,t)} u(x,y,t)\,dy,$$

$$\bar{u}_2(x,t) = \frac{1}{\bar{h}_2(x,t)} \int\limits_{\eta(x,t)}^{h_2} u(x,y,t)\,dy.$$

Condition (41) considered for non-constant velocities $\bar{u}_1$, $\bar{u}_2$ and non-constant layer thicknesses $\bar{h}_1$, $\bar{h}_2$ ($\bar{h}_1 + \bar{h}_2 = h_1 + h_2 = const$) provides hyperbolicity of hydrostatic two-layer shallow water equations

$$\bar{h}_{1t} + (\bar{u}_1 \bar{h}_1)_x = 0, \qquad \bar{h}_{2t} + (\bar{u}_2 \bar{h}_2)_x = 0,$$

$$\rho_1(\bar{u}_{1t} + \bar{u}_1 \bar{u}_{1x} + g\bar{h}_{1x}) = \rho_2(\bar{u}_{2t} + \bar{u}_2 \bar{u}_{2x} + g\bar{h}_{1x}).$$

In accordance with this non-linear formulation, layered flow can be locally unstable due to increased wave-induced shear even the constant upstream flow satisfies condition (41). This marginal instability of internal solitary waves was observed in laboratory experiments (Gavrilov 1994; Grue et al. 1999, 2000; Carr et al. 2008; Fructus et al. 2009), and studied theoretically (Choi et al. 2009; Almgren et al. 2012). Mathematical models of layered stable – unstable flows, which account also the non-linear dispersion of the waves and non-hydrostatic effects, they are currently under intense discussion in (Barros & Choi 2013; Lannes & Ming 2015; Duchene et al. 2016; Liapidevskii et al. 2018; Makarenko et al. 2018). However, the hydrostatic condition (41) seems to be appropriate by estimation of the stability limit with non-dimensional parameters from Section 3. Hence, we obtain the stability domain

$$|\sqrt{r}F_1 - F_2| < \sqrt{1+r}$$

which gives admissible set of the Froude points $(F_1, F_2)$ defined for the instant ratio of *local* layer depths $r = \bar{h}_1/\bar{h}_2$ and *local* Froude numbers in the solitary wave type flow described by stationary solution (39). This domain is shown for $r = 3$ in Fig. 2 (right panel) as unshadowed domain in the selected part of quarter-plane $(F_1, F_2)$.

## 7 Waves in Abyssal Shear Flows

We present in this Section a comparison of solutions of suggested mathematical model with the field data measured for internal solitary waves in weakly stratified abyssal currents. Figures 3 and 4 demonstrate fragments of temperature distribution in a quasi-steady shear bottom flow recorded from mooring station with a 350-m line of thermistors located over a depth of 4720 m at the entrance to the Romanche Fracture Zone in the equatorial Atlantic (van Haren et al. 2014). Trains of short-period (20÷30 min) internal waves modulated by tide propagate here along a sharp interface corresponding to the $0.85°C$ isotherm which separates the lower layer of cold Antarctic Bottom Water (AABW) from the overlying warmer AABW layer with temperature $\theta < 2.0°C$. It is seen from the figures that moored temperature data show permanently marginal stability of the flow with the Richardson numbers $0.25 < \mathrm{Ri} < 1$. Tidal amplification of the shear triggers the formation of small-scale overturns which create long trains of the Kelvin — Helmholtz billows. Bold curves in Fig. 3 and Fig. 4 show overlapped profiles of internal waves calculated from (39). Figure 3 demonstrates front-like wave (internal bore) affected by moderate small-scale shear instability. The non-dimensional amplitude of this wave is $a = \eta_{max}/h_1 = 0.16$, and calculated solution corresponds to the Froude point A with coordinates $(F_1, F_2) = (1.719, 0.891)$ shown in Fig. 2 (right panel). Point A belongs to the bore diagram which is tangential to the spectrum boundary at the point O corresponding to the vanishing amplitude limit $a = 0$. Figure 4 demonstrates the train of three subsequent waves with intense overturns distributed along with gently sloping wave tops. The amplitude of the leading wave shown in Fig. 4 is clearly lower than the amplitudes of the following two waves, which occur randomly and are almost identical. The solitary-wave type of these two waves corresponds to Froude point B with coordinates $(F_1, F_2) = (1.634, 0.959)$.

The upstream parameters used in the calculation were chosen from CTD and LADCP data of density and currents measured immediately at the fronts of selected waves. Undisturbed depth of lower layer $h_1$ was fixed by the equilibrium level of isotherm $0.85°C$ recorded at the buoy station shortly before the arrival of the large wave. In fact, this level changed gradually due to the tide, so that the value of $h_1$ varied within a day in the range from 170 m to 240 m. The depth of the upper layer $h_2$ was evaluated by fixing appropriate overlying isotherm, which remained almost horizontal; hence, this depth can be considered as the "rigid lid" in the model of the

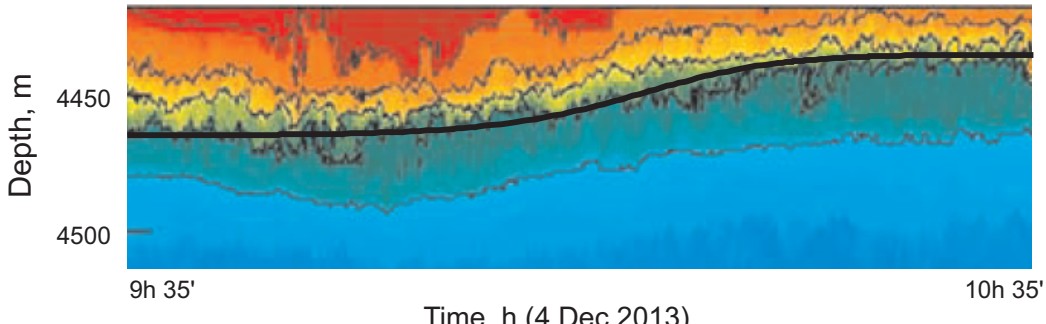

**Figure 3.** Internal front in abyssal stratified flow

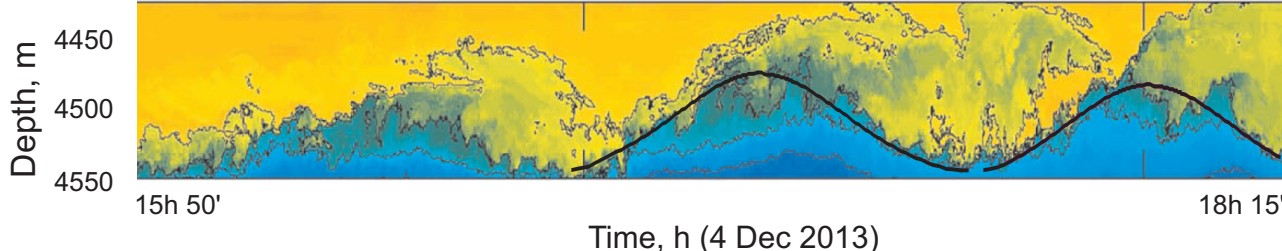

**Figure 4.** Train of interfacial solitary waves affected by the Kelvin — Helmholtz instability

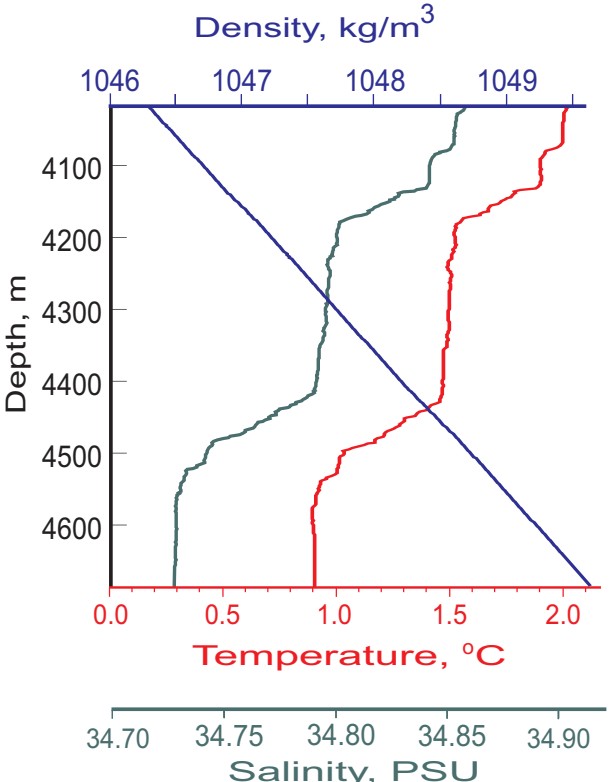

**Figure 5.** Profiles of the density, salinity and temperature

2.5-layer flow. This kind of "rigid" streamlines appears frequently in abyssal shear flows due to the formation of critical layers which separate clearly the currents with different thermohaline parameters (Morozov et al. 2012). The estimated depth $h_2$ also depends on the tidal activity, and this value varied in a wider range from 70 m to 350 m. Nevertheless, the resulting total depth $h = h_1 + h_2$ did not exceed the thickness of the entire AABW layer confined below a depth of 4000 m.

Upstream velocities $u_1$, $u_2$ were assumed as $u_1 = 35 \div 45\ cm/s$ and $u_2 = 10 \div 20\ cm/s$. These values correspond to the mean velocities measured in the western part of the Romanche Fracture Zone by a velocity LADCP profiler in the cruises of the R/V "Akademik Sergey Vavilov" (Tarakanov et al. 2013), and measured on the mooring station (van Haren et al. 2014). Maximal velocity of the flow in the lower layer, recorded by the mooring station within 6 months is as high as $65\ cm/s$. We consider that intense shear is induced here by the permanent inflow of bottom waters to the fracture zone through a narrow gap in its southern wall. Figure 5 demonstrates vertical profiles of temperature, salinity, and density measured on the shipborne mooring over a depth of 4760 m downstream the location of the bottom station. This thermohaline data highlight the layering of near-bottom flow with very smooth pycnoclines thickened due to intense mixing. In addition, the slope of almost linear density profile shown in Fig. 5 permits to evaluate the Boussinesq parameter $\sigma$ by the value $\sigma = 0.0027$. The Froude points $(F_1, F_2)$ related to these parameters match adequately with the range of internal solitary waves shown in the right panel of Fig. 2 (narrow

band marked by the orange color). The narrowness of this range reveals anomalously low amplitude dispersion of solitary waves which agrees with the field observations.

## 8 Conclusions

In this paper we have considered the problem of internal stationary waves at the interface between exponentially stratified fluid layers. We demonstrated that the non-linear DJL-model of weakly stratified 2.5-layer fluid flow can be reduced explicitly to an approximate non-linear ordinary differential equation describing large amplitude internal solitary waves. Parametric range of solitary waves is described, including regimes of broad plateau-shaped solitary waves and internal fronts. These wave regimes can be affected by the Kelvin — Helmholtz instability induced by the velocity shear at the interface; hence the marginal stability of internal waves could explain formation mechanism of a very long billow trains observed in the Romanche Fracture Zone.

*Acknowledgements.* This work was supported by the Russian Foundation for Basic Research (grants No 15-01-03942, 17-08-00085, 18-01-00648) and Interdisciplinary Program II.1 of SB RAS (project No 2). Roman Tarakanov was supported by the Russian Science Foundation (grant No 16-17-10149).

## Appendix A   Modal functions of linearized problem

Eigenfunction $W(y)$ considered in the strip $-\pi < y < 0$ has the form

$$W = a_1 e^{\frac{1}{2}\sigma_1 y} \begin{cases} e^{\varkappa_1 \pi} \dfrac{\sinh \varkappa_1(\pi+y)}{\varkappa_1} & (\lambda_1^2 < k_1^2 + \frac{\pi^2}{4}\sigma_1^2) \\ \pi + y & (\lambda_1^2 = k_1^2 + \frac{\pi^2}{4}\sigma_1^2) \\ \dfrac{\sin \varkappa_1(\pi+y)}{\varkappa_1} & (\lambda_1^2 > k_1^2 + \frac{\pi^2}{4}\sigma_1^2). \end{cases}$$

Similarly, eigenfunction $W(y)$ defined in upper layer which corresponds to the strip $0 < y < \pi/r$ has the form

$$W = a_2 e^{\frac{1}{2}\sigma_2 y} \begin{cases} e^{\varkappa_2 \pi} \dfrac{\sinh \varkappa_2(\pi-ry)}{\varkappa_2} & (\lambda_2^2 < k_2^2 + \frac{\pi^2}{4}\sigma_2^2) \\ \pi - ry & (\lambda_2^2 = k_2^2 + \frac{\pi^2}{4}\sigma_2^2) \\ \dfrac{\sin s_1(\pi-ry)}{\varkappa_2} & (\lambda_2^2 > k_2^2 + \frac{\pi^2}{4}\sigma_2^2). \end{cases}$$

The dimensionless wave-numbers $\varkappa_j$ ($j = 1, 2$) are introduced in formula (26), and the factors $a_j$ are the amplitude parameters.

## Appendix B   Dispersive term of the long-wave expansion

The coefficient $\psi^{(1)}(\xi, y)$ which gives the correction due to the dispersion in power expansion (35) has the form

$$\psi^{(1)} = \frac{\eta(\eta-y)}{2} \frac{\sin \alpha_1(y)}{\sin \alpha_1(\eta)} + \frac{\sin \alpha_1(y)}{2\lambda_1} \left( \frac{\eta}{\sin \alpha_1(\eta)} \right)_{\xi\xi} \times$$

$$\times \left\{ (\pi+\eta)\cot \alpha_1(\eta) - (\pi+y)\cot \alpha_1(y) \right\} + \frac{\eta^2}{6} \times$$

$$\times \left\{ \frac{\sin \lambda_1(y-\eta) - \sin \alpha_1(y)}{\sin^3 \alpha_1(\eta)} + \frac{1+\sin^2 \alpha_1(y)}{\sin^2 \alpha_1(\eta)} - \frac{\sin \alpha_1(y)}{\sin \alpha_1(\eta)} \right\},$$

in the lower layer $-\pi < y < \eta$. Similarly, we have

$$\psi^{(1)} = \frac{r^2 \eta(\eta-y)}{2} \frac{\sin \alpha_2(y)}{\sin \alpha_2(\eta)} + \frac{\sin \alpha_2(y)}{2\lambda_2} \left( \frac{\eta}{\sin \alpha_2(\eta)} \right)_{\xi\xi} \times$$

$$\times \left\{ (y-\pi/r)\cot \alpha_2(y) - (\eta-\pi/r)\cot \alpha_2(\eta) \right\} + \frac{r^2 \eta^2}{6} \times$$

$$\times \left\{ \frac{\sin \lambda_2 r(\eta-y) - \sin \alpha_2(y)}{\sin^3 \alpha_2(\eta)} + \frac{1+\sin^2 \alpha_2(y)}{\sin^2 \alpha_2(\eta)} - \frac{\sin \alpha_2(y)}{\sin \alpha_2(\eta)} \right\}$$

in the upper layer $\eta < y < \pi/r$. Here the functions $\alpha_j$ are given by the formula (36).

## Appendix C   Denominator of the non-linear long-wave equation

Denominator $Q$ in (37) has the form

$$2Q(\eta; F_1, F_2) =$$

$$\left( \pi F_1^2 - 2\sqrt{\pi} F_1 \eta \cot \alpha_1(\eta) + \eta^2 \cot^2 \alpha_1(\eta) \right) \times$$

$$\times \left( \frac{\eta+\pi}{\sin^2 \alpha_1(\eta)} - \sqrt{\pi} F_1 \cot \alpha_1(\eta) \right) +$$

$$+ \left( \frac{\pi F_2^2}{r^2} - 2\frac{\sqrt{\pi} F_2}{r} \eta \cot \alpha_2(\eta) + \eta^2 \cot^2 \alpha_2(\eta) \right) \times$$

$$\times \left( \frac{\pi - r\eta}{\sin^2 \alpha_2(\eta)} - \sqrt{\pi} F_2 \cot \alpha_2(\eta) \right)$$

where functions $\alpha_j(\eta)$ are given by formula (38) which is an approximate version of (36).

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
