# Peer review of "Internal waves in marginally stable abyssal stratified flows"

_Nonlinear Processes in Geophysics, 2018_

## Referee Comment (RC1) · Anonymous Referee #1 · 21 Feb 2018

The results of the research are new and interesting. They clarify one of the possible hydrodynamic effects of intense mixing of bottom waters in deep channels. The text is logically structured. Cumbersome formulas are given in the appendices. The manuscript corresponds to the theme of the special NPG issue dedicated to internal waves. The paper can be accepted and published after a few improvements.

Please also note the supplement to this comment:
https://www.nonlin-processes-geophys-discuss.net/npg-2018-1/npg-2018-1-RC1-supplement.pdf
* * *
[Figure]

**Supplement:**

**Referee Report**
of the manuscript by N. Makarenko, J. Maltseva, E. Morozov, R. Tarakanov. and K. Ivanova
"Internal waves in marginally stable abyssal stratified flows"

The authors suggest a mathematical model of nonlinear internal waves with dispersion in a fluid with two weakly stratified layers. The model is tested by comparing it with the field measurements. The novelty of the problem formulation is in the account for the peculiarities of the density stratification characteristics of abyssal currents. The key hypothesis applies the assumption that the small density jump of the fluid at the interface between two layers is of the same order of magnitude as the density gradients within the layers. Derivation of the model equation uses the long wave approximation, which is applied to the system of nonlinear equations for inviscid stratified fluid. Construction of the solution of the solitary wave type is reduced to quadrature. A chart of the wave regimes is constructed on the plane of densimetric Froud numbers. The authors conclude that the parametric domain of the solitary waves is close to the boundary of the Kelvin-Helmholtz instability zone of the main stream. This conclusion agrees qualitatively with the data of measurements of marginally stable flows of Antarctic Bottom Water in the abyssal fracture zones of the Atlantic, in which long series of strongly nonlinear internal waves with separation of the Kelvin-Helmholtz vortices are observed.

The results of the research are new and interesting. They clarify one of the possible hydrodynamic effects of intense mixing of bottom waters in deep channels. The text is logically structured. Cumbersome formulas are given in the appendices. The manuscript corresponds to the theme of the special NPG issue dedicated to internal waves. The paper can be accepted and published after a few improvements.

Remarks
1. Characteristic values of the Boussinesq parameters for abyssal currents that the authors use for comparison would complement and improve the text if they are added to the text. They will make the analysis more complete and characterize the domains of applicability of the new model.

2. Figure caption of Fig. 2 requires explanation for the right panel related to the indication of instability zone. There are comments in the text but it is highly desirable to include them into the figure caption. Many readers prefer analyzing the figures without referring to the text.

---

## Referee Comment (RC2) · Anonymous Referee #2 · 6 Apr 2018

Review of the article

**Internal waves in marginally stable abyssal stratified flows**

by *Nikolay Makarenko et al.*

In this paper, a strongly nonlinear long wave model is proposed to describe internal waves in a configuration that is known as 2.5 layers. The model prescribed extends the work by Voronovich (2003) by considering a background shear current. I particularly appreciate that the model is presented in its most general form, without recourse to the Boussinesq approximation, and the fact that the authors were able to present the dynamical system governing solitary-wave solutions in an explicit way as a first-order ODE for the interface displacement (the Hamiltonian of the system). Some solutions are then compared with in-situ measurements at the Romanche Fracture Zone in the equatorial Atlantic. I think its a valuable addition to the literature and I recommend publication pending the (minor) comments below.

- It is not clear to me how the the authors relate the background shear at infinity with the wave induced shear, known to trigger Kelvin-Helmholtz instabilities. Clearly this can be done if the wave under consideration is a front wave, but in the general case some good discussion would be welcome. Also, I do not see the relevance of considering the linear stability under long-wave perturbations, since in most two-layer models for any given shear, even if very small, there exists a wave number above which the problem is unstable. This issue has been the object of recent efforts how to regularize such models (see e.g. Choi et al. 2009; Lannes & Ming 2015; Duchene et al. 2016).

- I would have liked finding a more detailed study of the homoclinic orbits for the Hamiltonian system. In particular, higher-mode ISW could have been presented and conditions for the existence of trapped cores investigated. Could the latter be related with the singularities of $\mathrm{d}\eta/\mathrm{d}x$?

- For self-containedness, I would have preferred to find in §6 the density stratification and values for the background shear flow. It is not clear how the points $A$ and $B$ were determined to match the oceanic conditions. Also, aren't these points outside the blue region characterizing the mode-1 waves? Is the point $O$ located along the line $F_1 = F_2$? Is the solution on Fig. 4 a periodic wave?

---

## Author Comment (AC1) · 17 May 2018

1) Characteristic values of the Boussinesq parameters for abyssal currents that the authors use for comparison would complement and improve the text if they are added to the text. They will make the analysis more complete and characterize the domains of applicability of the new model.

The typical values of stratification parameters are discussed in the comments to Figure 5, which was specially added to the text. The Boussinesq parameter sigma corresponding to the density profile shown in Fig. 5 is sigma=0.0027, which provides applicability of the suggested theoretical model.

2) Figure caption of Fig. 2 requires explanation for the right panel related to the indi-

cation of instability zone. There are comments in the text but it is highly desirable to include them into the figure caption. Many readers prefer analyzing the figures without referring to the text.

The Kelvin-Helmholtz instability zone and the parametric region of the existence of solitary waves are now indicated in Fig. 2 (right panel).
* * *

---

## Author Comment (AC2) · 17 May 2018

1) It is not clear to me how the authors relate the background shear at infinity with the wave induced shear, known to trigger Kelvin-Helmholtz instabilities. Clearly this can be done if the wave under consideration is a front wave, but in the general case some good discussion would be welcome. Also, I do not see the relevance of considering the linear stability under long-wave perturbations, since in most two-layer models for any given shear, even if very small, there exists a wave number above which the problem is unstable. This issue has been the object of recent efforts how to regularize such models (see e.g. Choi et al. 2009; Lannes & Ming 2015; Duchene et al. 2016).

We determined the background shears for each individual wave in the wave packet of

soliton-shaped waves, which rapidly reach the parallel shear of the current. This has been done on the basis of the parameters of the current immediately before the wave. This hypothesis appeared applicable because the data of measurements agree well with the chart of wave regimes in the theoretical model and the calculated profiles of wave qualitatively correspond to the observed profiles. We can consider that the wave is a front wave because it exists in a narrow channel approximately 5 km wide and 300 m deep.

In order to determine the boundary of the parametric region of the Kelvin-Helmholtz instability we used the long-wave criterion, which is formed similarly both in the linear and non-linear versions. This important property allows us to trace the "dangerous" increase in the velocity shear induced by the solitary wave. It is clear that small-scale shortwave stability has been always observed in the measurements; however, it did not induce intense wave breaking. Strong mixing started exactly at the loss of stability, which was estimated precisely from the long-wave criterion of nonlinear hyperbolicity.

Following the reviewers comments we added detailed comments and formulas to the text, and also added more references.

2) I would have liked finding a more detailed study of the homoclinic orbits for the Hamiltonian system. In particular, higher-mode ISW could have been presented and conditions for the existence of trapped cores investigated. Could the latter be related with the singularities of dïAĺ/dx?

The authors agree that the problem on solitary waves of the higher modes and trapped vortex cores is extremely important in many applications and requires detailed research. This goal is beyond the scope of the problem reported in our manuscript. We would like to limit this publication by the analysis of the parameters of the first-mode solitary waves only, because it is strictly related to the interpretation of the data of our field studies in the Romanche Fracture Zone. In our numerical simulations we did not obtain solutions with singularities of dïAĺ/dx, although we do not exclude the existence

of such solutions for the constructed multi-parametric model. We thank the reviewer for the valuable comment and of course we plan to continue studying the qualitative properties of the solution of this Hamiltonian system.

3) For self-containedness, I would have preferred to find in para6 the density stratification and values for the background shear flow. It is not clear how the points A and B were determined to match the oceanic conditions. Also, aren't these points outside the blue region characterizing the mode-1 waves? Is the point O located along the line F1 = F2? Is the solution on Fig. 4 a periodic wave?

We significantly widened paragraph 6, and divided it into two parts. Paragraph 7 presents now the information about the density stratification and velocity. The mean velocity shear is approximately 15 cm sˆ(-1)/150 m, but it can exceed this value twice. The observed Richardson numbers are critical ∼0.25 as reported in [van Haren et al., 2014]; hence instability was revealed in the field observations. We consider that the shear is induced by the permanent inflow of bottom waters to the fracture zone through a narrow gap in its southern wall.

We added new Figure 5, which illustrates the characteristic profiles of density, temperature, and salinity in the region of our measurements. We also describe the method of determining the Froude points (F1, F2) on the basis of the field data.

The chart of wave regimes in the right panel of Fig. 2 is now given in improved colors for better visualization. The blue color shows the subcritical domain of the existence of linear waves of the first mode. The adjacent region of solitary waves of the first mode is supercritical; this narrow band is colored orange. In its turn it is bounded by the diagram of smooth bores (internal fronts), which is tangential to the spectra of linear waves at point O. Point O depends on the ratio of the depths of layers r and not necessarily belongs to the straight line F1 = F2.

The solution shown in Fig. 4 is non-periodic. The bottom station recorded irregular sequence of solitary internal waves with anomalously low amplitude dispersion. Two

very similar adjacent waves were occasionally noticed here. The wave before them has significantly smaller amplitude.
* * *

---

## Referee Report (RR1)

Final Referee Report

of the manuscript by N. Makarenko, J. Maltseva, E. Morozov, R. Tarakanov. and K. Ivanova
"Internal waves in marginally stable abyssal stratified flows"

The authors adequately responded to all comments.
I recommend to publication.